# Effect of smoking cessation interventions on abstinence and tuberculosis treatment outcomes among newly diagnosed patients: a randomized controlled trial

Muhammad Tahir Khan,[1] Sidra Zaheer,[1] Washdev Amar,[2] Kashif Shafique[1]

**ABSTRACT** The study evaluates the effectiveness of smoking cessation interventions [Behavioral Change Communication (BCC) and Behavioral Change Communication plus bupropion (BCC+)] compared to conventional Directly Observed Therapy Short Course (DOT) treatment in improving pulmonary tuberculosis treatment outcomes and abstinence among newly diagnosed pulmonary tuberculosis (PTB) patients, highlighting the scarcity of robust experimental studies. The current randomized controlled trial, conducted at Ojha Institute of Chest Diseases between October 2017 and June 2019, randomized 292 patients who were current smokers with newly diagnosed pulmonary tuberculosis into three arms: control ($n = 97$), BCC ($n = 97$), and BCC+ ($n = 98$) arms. The outcomes of the interventions were compared in terms of favorable treatment outcomes and abstinence achieved at the end of 6 months. Baseline characteristics were compared between groups. Cox regression quantified the effect size of interventions for both outcome variables and reported as (crude and adjusted) hazard ratios with 95% confidence intervals (CI). No statistically significant difference was observed in baseline characteristics in each arm. Both BCC+ and BCC showed a statistically significant effect in achieving favorable PTB outcomes at 6 months (aHR 2.37, 95% CI 1.52–3.70 and aHR 2.34, 95% CI 1.51–3.60), as well as for abstinence from smoking at 6 months (BCC+: aHR 4.03, 95% CI 2.18–7.44 and BCC: aHR 3.87, 95% CI 2.12–7.05) compared to the control arm. Both BCC and BCC+ aided by pharmacologic agents such as bupropion when incorporated with conventional DOTs were found to be significantly effective in attaining favorable tuberculosis treatment outcomes as well as in attaining smoking abstinence at the end of the 6-month treatment.

**IMPORTANCE** This study shows that adding smoking cessation programs (with or without extra drugs like bupropion) to standard Directly Observed Treatment Short Course (DOTs) treatment for people who have recently been diagnosed with pulmonary tuberculosis has a great positive impact on how well the overall antituberculosis treatment works. Our trial shows very promising results for such a combined therapy (DOTs and smoking cessation) in a country where the burden of both tuberculosis and smoking is very high.

**KEYWORDS** smoking cessation interventions, behavioral change communication, tuberculosis, smoking treatment outcomes

Despite the long-established association between cigarette smoking (CS) and pulmonary tuberculosis (PTB), very few scientific investigations have been undertaken so far to estimate the effect of smoking cessation interventions (SCIs) on PTB treatment outcomes (1, 2). However, this lack of published evidence regarding the importance of such interventions on 'PTB treatment outcomes' was only highlighted in

Address correspondence to Muhammad Tahir Khan, tahir.khan@duhs.edu.pk.

The authors declare no conflict of interest.

2016 when Cochrane published an "empty review" (3) concluding that there were no "individually randomized controlled trials" (RCTs) assessing the efficacy of SCIs on PTB treatment outcomes fulfilling the robust design of an RCT. However, scientific evidence regarding the morbidities caused by CS among PTB patients is well-established and shows the association between smoking and many of the adverse effects among PTB patients, like increased infectivity, (4) longer treatment duration, (5) loss of follow-up, (6) morbidity, (7–9) and even mortality (10, 11).

Unfortunately, the published few research studies exploring the effect of SCIs on PTB treatment outcomes from high-TB burden countries (1, 2, 12) have considerable methodological limitations. One of these studies, a cluster randomized trial (CRT), rather than an individually randomized controlled trial, which is methodologically better (13), assessed the effect of SCIs on PTB treatment outcomes and reported "no effect of SCIs on positive treatment outcomes." (2) Yet another study simply compared the differential efficacy of nicotine replacement therapy (NRT) on PTB treatment outcomes without comparing it with conventional anti-tuberculosis therapy (ATT) alone (1) (control group). ElSony A et al. (12) in their study, which was one of the first to assess the effect of SCI (in the form of brief advice), reported that the effect size of the SCI was 83% favorable TB treatment outcomes versus 59% with no SCI. Therefore, we found no individually randomized controlled trial assessing the effect of SCIs on "PTB treatment outcomes" in terms of standardized TB treatment outcomes. Consequently, the evidence generated by available studies on PTB treatment outcomes remains ambiguous and contradictory (2, 12). In addition, other related studies have simply considered "smoking abstinence" among PTB patients as their primary outcome variable, (14) and one observational study reported ATT outcomes among smoker and non-smoker PTB patient cohorts. (15)

Although all these studies well-served their defined objectives, during the literature search, we conducted a systematic review and meta-analysis to observe the pooled effects of interventional studies on abstinence (our second outcome variable) due to SCIs among PTB patients (since there was a lack of enough studies on PTB outcomes as a result of SCIs). We used keywords *smok\** and *tuberculosis* and limited the studies conducted among humans using Ovid in databases MEDLINE, EMBASE, and Cochrane Library for studies mentioning smoking and tuberculosis published between 1947 and 21 November 2016 (supplementary file: S1a).

We limited our review to only primary studies published in English, in which intervention was given for smoking cessation to TB patients. In order to calculate the effect size of smoking cessation interventions on TB treatment outcomes, we excluded studies that had no control group; hence, comparison was not possible. The study attrition diagram for the systematic review is shown as a supplementary file (supplementary file: S1b). A total of nine studies on smoking cessation interventions in TB patients were identified from different regions including Sudan (12), Indonesia (16, 17), Malaysia (18), China (19), Nepal (20), Bangladesh (21), Iran (22), and Pakistan (23). However, only five studies (12, 18, 20, 22, 23) fulfilled the final selection criteria for meta-analysis. In order to estimate the pooled effect of these interventional studies in achieving abstinence, meta-analysis was conducted that showed a pooled relative risk (RR) for abstinence of 5.66 (95% CI 3.75–8.55) (supplementary file: S1c).

Perceivably, an RCT assessing the efficacy of smoking cessation interventions on treatment outcomes poses multiple administrative and managerial challenges, due to long duration of treatment (ATT) and hence long follow-up, under-reporting of current and follow-up smoking statuses by patients, and adherence to treatment. Although all these challenges are likely to be encountered in any RCT, the population of interest in this particular case has somewhat unique characteristics, such as the majority of patients belonged to the low socio-economic stratum, low education, impermanent residence, and, great variations in smoking habits (24), making it harder for scientists to undertake and interpret the results of such a trial.

The objective of the current study was to assess the efficacy of SCIs in the form of behavioral change communication (BCC arm) alone and with BCC plus bupropion

(BCC+ arm) compared with the control arm (conventional DOT treatment alone) on PTB treatment outcomes among newly diagnosed PTB patients.

## MATERIALS AND METHODS

### Study design

This RCT was conducted at Dr. Iqbal Yaad Chest Clinic and Nazimabad Chest Clinic (OICD). During the study period (2017 to 2019), centers catered 8,973 presumptive TB cases, which detected and registered 1,451 smear-positive PTB cases alone.

### Participants

The RCT initially screened a total of 550 newly diagnosed pulmonary tuberculosis patients (both smear-positive or smear-negative) between October 2017 and June 2019 at two centers and enrolled 302 participants after obtaining written informed consent from participants (which was available in English, Urdu, Sindhi, and Pushto with a detailed explanation of the study protocol). The principal investigator obtained the informed consent (with the help of translators where required). The enrollment in the RCT was done as per the inclusion criteria of (i) being registered under Directly Observed Treatment Shortcourse (DOTs) at OICD, (ii) age ≥18 years, and (iii) being reported to have smoked at least 100 cigarettes and being a "current smoker." Current smoking status was determined using self-reporting and supported by the expired carbon monoxide (eCO) level of ≥7 ppm using piCO Smokerlyzer (Bedfont® Scientific Ltd.) carbon monoxide breath test monitor (as per Bedfont Scientific Ltd. device protocol) and serum cotinine level of ≥10 ng/mL at baseline (25) simultaneously. The screened PTB patients were then labeled under either of three following categories as per the Center for Disease Control and Prevention (CDC): (26) (i) Current smoker: an adult who has smoked 100 cigarettes/bidi in his or her lifetime and who currently smokes cigarettes/bidi. (27) (ii) Former smoker: an adult who has smoked at least 100 cigarettes/bidi in his or her lifetime but who had quit smoking at the time of interview. (27) (iii) Never smoker: an adult who has never smoked or who has smoked less than 100 cigarettes/bidi in his or her lifetime. (27)

Patients were excluded based on the following criteria: (i) extra-pulmonary tuberculosis (EPTB) in addition to pulmonary tuberculosis (PTB), (ii) having drug-resistant tuberculosis (DR-TB), (iii) pregnancy, (iv) permanent residence outside Karachi, (v) known human immune deficiency virus (HIV)-positive status, (vi) advanced liver disease, and (vii) history of epilepsy [as an intervention arm contained bupropion SR (Butrin SR, Genetics Pharmaceuticals), which is contraindicated in epilepsy]. (28)

Of the total 302 patients fulfilling the inclusion criteria, six refused to continue participating in the trial (during the first week due to personal reasons), one was excluded on the basis of epilepsy history, two were excluded due to DR-TB (based on the results of GeneXpert), while one was excluded due to permanent residence outside Karachi.

### Sample size

The sample size was calculated using the efficacy of the SCI on PTB patients' treatment outcomes in "Intervention TB treatment clinics" as 83% versus 59% in "Control Centers" (where no smoking cessation intervention was given) by El Sony et al. (12) The minimum required sample size was calculated to be 218 at 99% confidence level and 90% power. Keeping in consideration the potential attrition among PTB patients undergoing a smoking cessation intervention, the sample size was increased by 12% to 240 (80 in each arm), based on the attrition results reported by Bam TS et al. (16) However, a total of 292 PTB patients were consecutively enrolled in the trial, and the final analysis was done on a total of 292 study participants (97 in the control arm, 97 in the BCC arm, and 98 in the BCC+ arm).

## Randomization protocol and blinding

Remote randomization was carried out, by a centralized mechanism (based at School of Public Health, DUHS), where every enrolled participant was coded, and the information was sent by the focal person at enrollment centers, responsible for the facilitation of the study participants' recruitment. The information was sent to a dedicated cell phone line under the supervision of the principal investigator, and then random allocation of the intervention group was made with the help of a pre-generated random allocation sequence using MS Excel. The staff at the OICD clinics and the laboratory personnel responsible for determining the AFB positivity/negativity were blinded to the allocated intervention and participation in the trial of the patients.

## Interventions

The interventions were divided into two groups, with a third group being the control. The details of the three groups are as follows:

1. **Behavioral Change Therapy alone (BCC)**, consisting of 10–15 minutes of BCC sessions (every month) focusing on reassurance regarding tuberculosis treatment and re-enforcement of smoking cessation based on the BCC toolkit designed for both males and females separately for the Pakistani population by Siddiqi K et al. with their permission (29), which, in turn, was designed in light of the WHO's 5 A approach toward smoking cessation. (30)
2. **Behavioral Change Therapy with Bupropion** consisted of the same BCC counseling sessions in addition to bupropion SR 150 mg twice daily for 10 weeks (in tablet form). Bupropion prescription and the assessment of the patients regarding any potential adverse effects were done by a clinical psychiatrist.
3. **Control arm** contained PTB smokers only receiving routine anti-tuberculosis treatment, with a one page leaflet regarding the hazards of smoking.

## Outcome determination

The main outcome of interest was the PTB treatment outcome at the end of 24 weeks (6 months) of ATT, categorized into six mutually exclusive classes as defined by the WHO Global TB report 2016. (31) The six main outcomes were (i) *Cured*, (ii) *Treatment completed*, (iii) *Treatment failure*, (iv) *Death*, (v) *Lost to follow-up,* and (vi) *Not evaluated*. The assessment for these outcomes was done using RCT follow-up, and the findings were compared with TB03 registers for any discrepancies. The categories, *Cured* and *Treatment Completed* were clumped together as "successful treatment outcome" while *Treatment failure*, *Death*, and *Lost to follow-up* and *Not evaluated* were grouped as "unsuccessful treatment outcome." The secondary outcome of interest was "abstinence," which was also calculated as the main outcome variable as staying away from smoking, and it was assessed at week 1, week 4, week 12, and week 24 using (i) self-reported abstinence using questionnaire and (ii) expired carbon monoxide (eCO) levels using a Micro Smokerlyzer (Bedfont Scientific) of less than 7 ppm. Patients falling into categories of "former smokers" and "never smokers" were excluded from the study as per exclusion criteria.

Two deaths occurred during the RCT, one in the control group (male aged 60 years) and the other in the BCC group (male aged 75 years), due to natural causes after 4 months of (median) follow-up. We decided not to exclude their data from the analyses as removing them from the analyses did not cause any significant difference in the comparison treatment outcome between intervention groups.

## Baseline covariates

### Baseline bacterial load

This was assessed as per the diagnostic protocol of the National Tuberculosis Control Program (NTP) at the time of diagnosis using sputum smear microscopy by using the count of acid-fast bacilli (AFB, bacteria of tuberculosis) and was categorized into the following categories:

Scanty: (1–9 acid-fast bacilli in 100 fields of microscopy)
1+: (10–99 acid-fast bacilli in 100 fields of microscopy)
2+: (1–9 acid-fast bacilli in 100 fields of microscopy)
3+: (10 + acid-fast bacilli in 100 fields of microscopy)

### Age of smoking onset

Age of smoking onset was recorded by asking "what was your age when you first started smoking regularly?" and the response was recorded in *years of age*.

### Number of packs-years

Number of pack-years was also calculated using the following formula:

$$\text{Number of pack} - \text{years} = (\text{number of cigarettes smoked per day}/20) \times \text{number of years}$$

### Baseline motivation to quit

Baseline motivation to quit was determined by asking the following questions, all measured on a Likert scale, (i) "What is your opinion on quitting cigarette smoking?" (ii) "How important for you it is now that you should quit smoking?" (iii) "How determined you are this time to quit smoking?" and (iv) "I am confident that this time if I decide to quit then I can permanently abstain from smoking." The final motivation score was calculated by adding the Likert scale scores (where a score of 1 showed least motivation and that of 10 showed highest motivation) and was categorized as follows based on the total motivation score:

11–12: Very Strong (strongest motivation to quit smoking at the start of the study)
9–10: Moderately Strong (moderately strong motivation to quit smoking at the start of the study)
7–8: Weak (weak motivation to quit smoking at the start of the study)
5–6: Very Weak (very weak motivation to quit smoking at the start of the study)
<4: No Motivation (no motivation to quit smoking at the start of the study)

### Wealth index

Wealth index was calculated using principal component analysis using the asset variables (i) "fuel used for cooking," (ii) "job," (iii) "house type," (iv) "house ownership," (v) "room density (number of persons sleeping in same room)," (vi) "flush (toilet)," (vii) "landline telephone," (viii) "television," (ix) "fridge," (x) "car," and (xi) motorcycle. Selection of variables was based on the rule of thumb that those assets were excluded that were owned by more than 95% or less than 5% of the study population, like mobile phone. All these variables were transformed into bivariate variables. The wealth index was categorized into quintiles, where the first quintile was richest and fifth was poorest.

### Nicotine dependence

Baseline nicotine dependence was quantified using Fagerström Test for Nicotine Dependence (FTND) (32), and the participants were classified according to the following scoring system:

1–2: Low Dependence (low dependence on nicotine at the start of the study)

3–4: Low-to-Moderate Dependence (low-to-moderate dependence on nicotine at the start of the study)

5–7: Moderate Dependence (moderate dependence on nicotine at the start of the study)

≥8: High dependence (high dependence on nicotine at the start of the study)

For the sake of statistical analyses, we merged the first two categories as "minimally dependent," "moderately dependent," and last category as "highly dependent."

## Intervention and Follow-up

Interventions were carried out under the principal investigator and the intervention arm 2 BCC+ involved a psychiatrist (co-investigator) who assessed the study participants at baseline and was responsible for the prescription of bupropion. Follow-ups were conducted at 1-month intervals.

## Statistical treatment

Data were initially entered, cleaned, and coded using MS Excel 2007 for MS Windows. Afterward, the analyses were done using Stata 14 for MS Windows. The details of statistical analyses are as follows.

### Descriptive analyses

From a total of 292 (97: control, 97: BCC, and 98: BCC+) patients, all patients were included in the analyses. Despite that there were two deaths among the trial participants, they were not excluded from the analyses as the deaths occurred after the median follow-up of 4 months, and we compared the analyses results including and excluding the two participants but found no statistically significant difference. Mean and standard deviations for all continuous variables and the chi-squared test (or Fisher exact test where warranted) for categorical variables were used to compare all three intervention groups to explore any statistically significant difference among any of the groups' baseline characteristics. Variables compared were age, sex, wealth index, smear positivity status of PTB, baseline bacterial load, age of smoking initiation, mean years of smoking, number of pack years smoked, expired carbon monoxide levels, nicotine dependence based on FTND scores, and motivation to quit smoking.

### Inferential analyses

Cox regression was employed in order to calculate relative risks with 95% confidence intervals (*CIs*) after clumping the six PTB treatment outcomes into two categories (i) favorable outcome and (ii) unfavorable, as mentioned in the *Methodology* section. Similarly, the second primary outcome, abstinence was also categorized into two, as attained at week 1 after intervention and at week 24 (completion of the trial). Univariate and multivariate Cox regression analyses were performed to assess independent predictors of a successful outcome (event). Multivariate hazard ratios were adjusted for variables that were reported in literature as clinical significant factors (33, 34). A *P*-value of <0.05 was considered statistically significant.

## RESULTS

The baseline characteristics of the three study groups are shown in Table 1. There was no statistically significant difference between baseline characteristics of study participants in all three study arms. Within all trial groups, 97/98 (98.9%) in the BCC+ group, 93/97 (95.8%) in the BCC group, and 88/97 (90.7%) in the control group had a favorable outcome at 6 months. The point abstinence rate from smoking was 53.1% at baseline, which increased to 89.7% in the BCC+ group after intervention. Similarly, the point

**TABLE 1** Comparison of baseline characteristics of pulmonary tuberculosis patients enrolled in the study according to three groups (n = 292)[a]

| Baseline characteristics | Control (n = 97) n (%) | BCC (n = 97) n (%) | BCC+ (n = 98) n (%) | P-value |
|---|---|---|---|---|
| Age (years) Mean (SD) | 38.50 (19.31) | 38.87 (20.38) | 35.33 (18.99) | 0.382 |
| Sex | | | | |
| Female | 6 (6.20%) | 5 (5.20%) | 4 (4.10%) | 0.801 |
| Male | 91 (93.80%) | 92 (94.80%) | 94 (95.90%) | |
| Wealth index (n = 145) | | | | |
| First quintile (wealthiest) | 17 (41.50%) | 17 (32.70%) | 21 (40.40%) | 0.464 |
| Second quintile | 5 (12.20%) | 10 (19.20%) | 5 (9.60%) | |
| Third quintile | 15 (36.60%) | 13 (25.00%) | 12 (23.10%) | |
| Fourth quintile | 2 (4.90%) | 7 (13.50%) | 8 (15.40%) | |
| Fifth quintile (poorest) | 2 (4.90%) | 5 (9.60%) | 6 (11.50%) | |
| Pulmonary tuberculosis | | | | |
| Smear positive | 71 (73.20%) | 73 (75.30%) | 69 (70.40%) | 0.746 |
| Smear negative | 26 (26.80%) | 24 (24.70%) | 29 (29.60%) | |
| Bacterial load (on AFB smear microscopy) (n = 213) | | | | |
| Scanty | 2 (2.80%) | 8 (11.00%) | 4 (5.80%) | 0.407 |
| 1+ | 20 (28.20%) | 16 (21.90%) | 17 (24.60%) | |
| 2+ | 23 (32.40%) | 17 (23.30%) | 22 (31.90%) | |
| 3+ | 26 (36.60%) | 32 (43.80%) | 26 (37.70%) | |
| Age of smoking initiation, Mean (SD) | 16.13 (4.60) | 16.08 (4.74) | 16.77 (4.70) | 0.510 |
| Mean years smoked, Mean (SD) | 22.08 (16.97) | 22.97 (18.16) | 18.32 (16.47) | 0.176 |
| Number of pack years, Mean (SD) | 18.32 (21.09) | 18.14 (21.57) | 14.66 (19.34) | 0.381 |
| Expired carbon monoxide, Mean (SD) | 12.28 (7.14) | 11.43 (6.48) | 12.24 (7.12) | 0.620 |
| FTND categories | | | | |
| Minimally dependent | 58 (59.80%) | 59 (60.80%) | 62 (63.30%) | 0.989 |
| moderately dependent | 30 (30.90%) | 30 (30.90%) | 28 (28.60%) | |
| highly dependent | 9 (9.30%) | 8 (8.20%) | 8 (8.20%) | |
| Motivation to quit | | | | |
| Very strong | 61 (62.90%) | 57 (58.80%) | 51 (52.00%) | 0.581 |
| Moderately strong | 20 (20.60%) | 28 (28.90%) | 24 (24.50%) | |
| Weak | 11 (11.30%) | 8 (8.20%) | 15 (15.30%) | |
| Very weak | 4 (4.10%) | 4 (4.10%) | 7 (7.10%) | |
| No motivation | 1 (1.00%) | 0 (0.00%) | 1 (1.00%) | |

[a]Parentheses values for age, age of smoking initiation, number of packs per year, and expired carbon monoxide level show standard deviation. P-value calculated using Kruskal–Wallis or χ test. FTND: Fagerström Test for Nicotine Dependence.

abstinence rate from smoking was higher in the BCC group (87.6%) than in the control group (30.9%) after intervention.

The multivariate adjusted model showed that the chance of a favorable PTB outcome was 2.37 times in the BCC+ group (aHR 2.37, 95% CI 1.52–3.70) and 2.34 times in the BCC group (aHR 2.34, 95% CI 1.51–3.60) compared to the controls after the follow-up. In terms of attaining abstinence from smoking at 6 months, the BCC+ and BCC groups have the chance of abstinence 4.03 times (aHR 4.03, 95% CI 2.18–7.44) and 3.87 times (aHR 3.87, 95% CI 2.12–7.05), respectively, compared with the control group. (Table 2) The Kaplan–Meier curve also showed that the higher successful rate of abstinence was achieved among participants in the BCC+ group compared to other trial groups. (Fig. 1)

## DISCUSSION

This randomized controlled trial found that both BCC and BCC+ are statistically significant SCIs in achieving favorable PTB outcomes as well as abstinence at 6 months of ATT (end of treatment) compared to control. The effect size of the interventions differed nominally in terms of the treatment outcome between BCC and BCC+ arms; however, it

TABLE 2 Univariate and multivariate Cox regression results of the efficacy of cessation interventions versus control for successful pulmonary tuberculosis treatment outcomes and abstinence (n = 292)

| Characteristics | Favorable PTB outcome at 6 months | | | | Point abstinence at 6 months | | | |
|---|---|---|---|---|---|---|---|---|
| | cHR (95% CI) | P-value | aHR (95% CI) | P-value | cHR (95% CI) | P-value | aHR (95% CI) | P-value |
| Trial groups | | | | | | | | |
| Controls | Ref. | | Ref. | | Ref. | | Ref. | |
| BCC | 2.18 (1.61–2.96) | <0.001 | 2.34 (1.51–3.60) | <0.001 | 3.71 (2.43–5.65) | <0.001 | 3.87 (2.12–7.05) | <0.001 |
| BCC+ | 2.49 (1.84–3.39) | <0.001 | 2.37 (1.52–3.70) | <0.001 | 4.07 (2.67–6.20) | <0.001 | 4.03 (2.18–7.44) | <0.001 |

[a]BCC: behavioral change communication; BCC+: behavioral change communication plus bupropion; PTB: pulmonary tuberculosis; abstinence: abstinence from smoking; cHR: crude hazards ratio; CI: confidence intervals; aHR: hazards ratio adjusted for all clinically and sociodemographically important variables; these included education in years, wealth index, anxiety and depression, smear-positive at baseline, motivation to quit, and nicotine dependency.

was marginally greater in the BCC+ arm compared to the BCC arm in terms of achieving abstinence.

Interventional studies from different parts of the world have mainly reported similar positive effects of the SCIs on abstinence from smoking among tuberculosis patients (1, 12, 14, 23, 35), supporting the findings of our study. These previously reported interventions among PTB patients mainly comprised counseling, ranging from brief advice (2, 12) to detailed BCC (1, 14, 23). However, some of the studies also included bupropion in addition to counseling, which showed inconclusive results related to the additional effect of bupropion (14, 23). Siddiqi et al. (23) and Aryanpur et al. (14) incorporated a second intervention arm (BCC plus bupropion) in their respective studies, where former authors

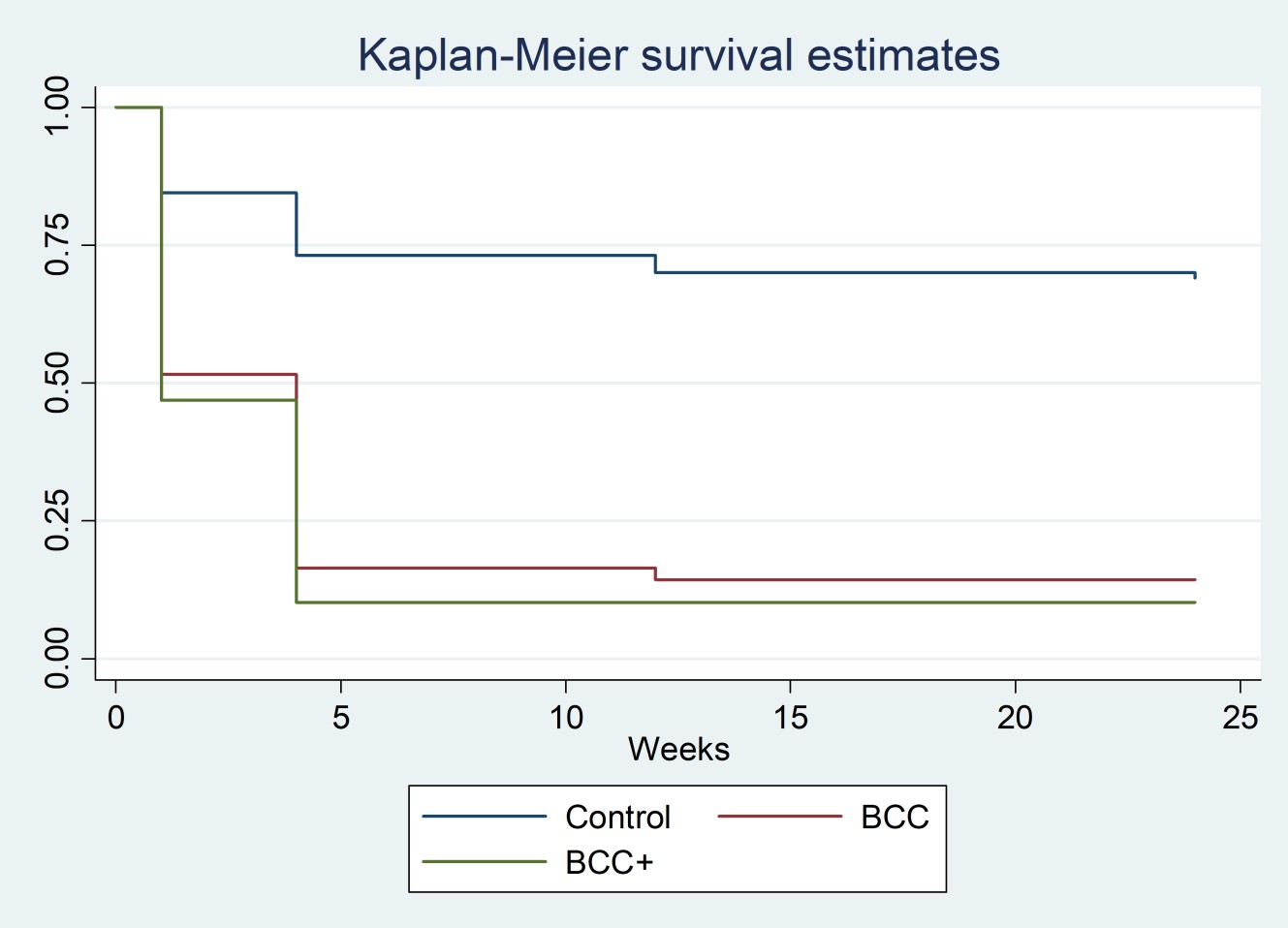

FIG 1 Kaplan–Meier estimates for continuous abstinence among study participants according to three study groups (n = 292).

reported no enhancement in the efficacy of BCC if given with bupropion compared to BCC alone in terms of smoking cessation, whereas the results of the latter reported significantly enhanced efficacy of the BCC plus bupropion group. This could have been due to an important methodological difference between the two studies that Siddiqi et al. (23) used BCC based on the WHO's 5As principle in both BCC and BCC plus bupropion groups, while Aryanpur et al.'s "brief advice alone method" in the BCC arm and robust WHO's 5As principle based BCC in BCC plus bupropion (14). The discrepancy between the counseling method used with the bupropion group and the brief advice alone group raised questions regarding the internal validity of the study results. However, we also found that both BCC alone and BCC in combination with bupropion were effective in attaining abstinence in newly diagnosed PTB patients. However, the BCC plus bupropion group had a stronger chance of attaining abstinence compared to BCC alone.

However, our study, the first individually randomized controlled trial, testing the efficacy of SCIs against no SCI as the control group, among newly diagnosed PTB patients, found that only BCC plus bupropion was effective in attaining favorable PTB treatment outcomes compared to the BCC alone and control groups. Sharma SK et al., (1) in their RCT, reported no statistically significant positive effect on PTB treatment outcomes when compared between the counseling arm and counseling plus nicotine replacement therapy (NRT; another form of SCI with a different mechanism of action compared to bupropion, which acts on the central nervous system by reducing the urge to smoke). This study also however did not have any "no intervention control arm," considering counseling alone as a "control arm," thus only comparing the differential effect of counseling alone and counseling plus NRT on PTB treatment outcomes. On the other hand, a cluster randomized trial (consisting of 156 patients from 17 clusters) from India (2) reported no effect of SCI on TB treatment outcomes. Our study thus answers the crucial question of whether SCIs have a positive impact on the TB treatment outcomes when observed through an individually randomized controlled trial with "no intervention" as the control arm. Counseling intervention in our study was based on the WHO's 5A technique and was especially designed and pretested and validated for the local population previously used by Siddiqi et al. (23)

This trial suggests that incorporating SCIs (either with or without an additional pharmacologic intervention like bupropion) within conventional DOT treatment for PTB patients has a very significant effect on the overall treatment outcomes as well as attaining abstinence among these patients in Pakistan if BCC is properly incorporated, a country where there is a double burden of high prevalence of PTB and smoking. Despite a relatively small number of observations in our trial, the results showed promising evidence.

## Conclusion

Stand-alone BCC or BCC combined with bupropion as a smoking cessation intervention when incorporated with conventional DOTs was found to be significantly effective in attaining favorable tuberculosis treatment outcomes as well as in attaining abstinence.

## ACKNOWLEDGMENTS

We acknowledged Prof. Kamran Siddiqi and his team at University of York, UK, for providing the pretested, validated behavioral change communication toolkit for the Pakistani population.

There was no external funding for this project.

M.T.K. conceived the research question, conducted all the literature reviews, data collection, and management, data analysis, and manuscript drafting and reviewing. W.A. worked as the Clinical Psychiatrist in the trial and was responsible for the evaluation of patients for exclusion of epilepsy and prescription of bupropion. S.Z. helped in the analysis and preparation of tables. K.S. supervised the research project and conducted

the final editing and review of the manuscript. All authors approved the final version of the manuscript.

## AUTHOR AFFILIATIONS

[1]School of Public Health, Dow University of Health Sciences, Karachi, Pakistan
[2]Dr. A. Q. Khan Institute of Behavioral Sciences, Karachi, Pakistan

## AUTHOR ORCIDs

Muhammad Tahir Khan ⓘ http://orcid.org/0000-0002-0281-7650

## DATA AVAILABILITY

Complete data are available upon request.

## ETHIC APPROVAL

The trial was registered with ID NCT04848246 at ClinicalTrials.gov, and the RCT obtained institutional review board (IRB) approval with registration no. IRB-898/DUHS/Approval/2017/136.

## ADDITIONAL FILES

The following material is available online.

### Supplemental Material

**Meta analysis (Spectrum03878-23-s0001.docx).** Forest plot showing the pooled relative risk of abstinence among TB patients with smoking cessation intervention compared to control group (no intervention) along with weights of individual studies
**Attrition diagram (Spectrum03878-23-s0002.docx).** Study attrition diagram for systematic review and meta analysis.
**Search strategy (Spectrum03878-23-s0003.docx).** Study search strategy using OVID for meta analysis.

### Open Peer Review

**PEER REVIEW HISTORY (review-history.pdf).** An accounting of the reviewer comments and feedback.

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
