## [Reviewer comments · Microbiology Spectrum]

Microbiology Spectrum

Effect of smoking cessation interventions on abstinence and tuberculosis treatment outcomes among newly diagnosed patients: A randomized controlled trial

Muhammad Tahir Khan, Sidra Zaheer, Washdev Amar, and Kashif Shafique

Corresponding Author(s): Muhammad Tahir Khan, Dow University of Health Sciences

Review Timeline:

Submission Date:	November 16, 2023
Editorial Decision:	December 27, 2023
Revision Received:	January 11, 2024
Editorial Decision:	January 25, 2024
Revision Received:	January 26, 2024
Accepted:	January 29, 2024

Editor: Sadjia Bekal

Reviewer(s): Disclosure of reviewer identity is with reference to reviewer comments included in decision letter(s). The following individuals involved in review of your submission have agreed to reveal their identity: Abraham Terna Lan (Reviewer #1)

Transaction Report:

DOI: <https://doi.org/10.1128/spectrum.03878-23>

Re: Spectrum03878-23 (Effect of smoking cessation interventions on abstinence and tuberculosis treatment outcomes among newly diagnosed patients: A randomized controlled trial)

Dear Dr. Muhammad Tahir Khan:

Thank you for the privilege of reviewing your work. Below you will find my comments, instructions from the Spectrum editorial office, and the reviewer comments.

This is an interesting and complete study. I have some minor suggestions.

1. Line 23: In purpose, authors are citing BCC and BCC+ that are finally explained in Paragraph (page 7, lines 158-169). It's important for readers to add a short description of BCC and BCC+ in purpose paragraph to help in understanding the study.
2. The same observation for DOTs
3. Lines: 193-196; Lines 211-215; Lines 228-231 : Authors should write these descriptions in in the same line, separated by semicolon or parentheses
4. Please give attention to references and their format according to journal policy. For example, references 25 and 35 are incomplete and not found by readers in their current format. A link will be helpful.

Revision Guidelines

Sincerely,
Sadjia Bekal

Editor
Microbiology Spectrum

Reviewer #1 (Comments for the Author):

The authors did a meticulously thorough job, highly commendable. Research protocols are clearly observed and carefully implemented. The research findings are clearly reported without ambiguity. However there seem to be a minor error on line 309 - 310 "...Similar insignificant effect of another SCI, nicotine replacement therapy (NRT), was reported by However Aryanpur et al, (14) reported that 'counseling plus bupropion' was more effective in attaining.) Kindly revisit this segment of the report for better understanding.

Effect of smoking cessation interventions on abstinence and tuberculosis treatment outcomes among newly diagnosed patients: A randomized controlled trial by Muhammad Tahir Khan et al., represents a product of quality research. The authors did a meticulously thorough job, highly commendable. Research protocols are clearly observed and carefully implemented. The research findings are clearly reported without ambiguity. However, there seem to be a minor error on line 309 - 310

"...Similar insignificant effect of another SCI, nicotine replacement therapy (NRT), was reported by However Aryanpur et al, (14) reported that 'counseling plus bupropion' was more effective in attaining."

Kindly revisit this segment of the report for better understanding.

Dear Editor,

Response to the Comments

Respected editorial staff and reviewer, thank you very much for your constructive feedback. Please find below the comments and their respective responses. The changes are highlighted.

I hope that the reviewers find the manuscript worthy of publication in your esteemed journal after these amendments.

Best Regards

Muhammad Tahir Rizwan Khan
School of Public Health, Dow University of Health Sciences, Karachi, Pakistan
ORCID: <https://orcid.org/0000-0002-0281-7650>
Tel +92 300 9257633
Email: tahir.khan@duhs.edu.pk

Comment 1. Line 23: In purpose, authors are citing BCC and BCC+ that are finally explained in Paragraph (page 7, lines 158-169). It's important for readers to add a short description of BCC and BCC+ in purpose paragraph to help in understanding the study.

Response: Short description has been added.

Comment 2. The same observation for DOTs

Response: Incorporated

Comment 3. Lines: 193-196; Lines 211-215; Lines 228-231 : Authors should write these descriptions in in the same line, separated by semicolon or parentheses

Response: Incorporated

Comment 4. Please give attention to references and their format according to journal policy. For example, references 25 and 35 are incomplete and not found by readers in their current format. A link will be helpful.

RESPONSE: The references are removed. Other references are also put according to the journal's referencing style (ASM Journals).

Reviewer's Comment: Reviewer #1 (Comments for the Author):

The authors did a meticulously thorough job, highly commendable. Research protocols are clearly observed and carefully implemented. The research findings are clearly reported without ambiguity. However there seem to be a minor error on line 309 - 310 "...Similar insignificant effect of another SCI, nicotine replacement therapy (NRT), was reported by However Aryanpur et al, (14) reported that 'counseling plus bupropion' was more effective in attaining.) Kindly revisit this segment of the report for better understanding.

RESPONSE: The part has been rewritten for clarity.

Re: Spectrum03878-23R1 (Effect of smoking cessation interventions on abstinence and tuberculosis treatment outcomes among newly diagnosed patients: A randomized controlled trial)

Dear Dr. Muhammad Tahir Khan:

Thank you for the privilege of reviewing your work. Below you will find my comments, instructions from the Spectrum editorial office, and the reviewer comments.

Revision Guidelines

Sincerely,
Sadjia Bekal
Editor
Microbiology Spectrum

Reviewer #1 (Comments for the Author):

I confirmed that the authors adequately addressed the concerns I raised during the initial review process

Effects of smoking cessation intervention on abstinence and tuberculosis treatment outcomes among newly diagnosed patients: A randomized controlled trial by Muhammed Tahir Khan et al., based on my previous comments and concerns raised I wish to confirmed that the authors satisfactorily attended to the issues I raised as shown below indicated in alphabetical order “A” and “B” to represent the review concerns and the corrections thereof respectively.

- A. However, there seem to be a minor error on line 309 - 310 "...Similar insignificant effect of another SCI, nicotine replacement therapy (NRT), was reported by However Aryanpur et al, (14) reported that 'counseling plus bupropion' was more effective in attaining...,
- B. has been modified to (line 305-311) “similar positive effects of the SCIs on abstinence from smoking among tuberculosis patients (1,12,14,23,35) supporting findings of our study. These previously reported interventions among PTB patients mainly comprised of counseling, ranging from brief advice (2,12) to detailed BCC. (1,14,23) However, some of the studies also included Bupropion in addition to counseling which showed inconclusive results related to the additional effect of Bupropion. (14,23) Siddiqi et al, (23) and Aryanpur et al (14) incorporated a second intervention arm (BCC plus bupropion) in their respective studies, “

Dear Editor,

Response to the Comments

Respected editorial staff and reviewer, thank you very much for your constructive feedback. Please find below the comments and their respective responses. The changes are highlighted.

I hope that the reviewers find the manuscript worthy of publication in your esteemed journal after these amendments.

Best Regards

Muhammad Tahir Rizwan Khan
School of Public Health, Dow University of Health Sciences, Karachi, Pakistan
ORCID: <https://orcid.org/0000-0002-0281-7650>
Tel +92 300 9257633
Email: tahir.khan@duhs.edu.pk

Comment 1. Line 23: In purpose, authors are citing BCC and BCC+ that are finally explained in Paragraph (page 7, lines 158-169). It's important for readers to add a short description of BCC and BCC+ in purpose paragraph to help in understanding the study.

Response: Short description has been added.

Comment 2. The same observation for DOTs

Response: Incorporated

Comment 3. Lines: 193-196; Lines 211-215; Lines 228-231 : Authors should write these descriptions in in the same line, separated by semicolon or parentheses

Response: Incorporated

Comment 4. Please give attention to references and their format according to journal policy. For example, references 25 and 35 are incomplete and not found by readers in their current format. A link will be helpful.

RESPONSE: The references are removed. Other references are also put according to the journal's referencing style (ASM Journals).

Reviewer's Comment: Reviewer #1 (Comments for the Author):

The authors did a meticulously thorough job, highly commendable. Research protocols are clearly observed and carefully implemented. The research findings are clearly reported without ambiguity. However there seem to be a minor error on line 309 - 310 "...Similar insignificant effect of another SCI, nicotine replacement therapy (NRT), was reported by However Aryanpur et al, (14) reported that 'counseling plus bupropion' was more effective in attaining.) Kindly revisit this segment of the report for better understanding.

RESPONSE: The part has been rewritten for clarity.

Re: Spectrum03878-23R2 (Effect of smoking cessation interventions on abstinence and tuberculosis treatment outcomes among newly diagnosed patients: A randomized controlled trial)

Dear Dr. Muhammad Tahir Khan:

Your manuscript has been accepted, and I am forwarding it to the ASM production staff for publication. Your paper will first be checked to make sure all elements meet the technical requirements. ASM staff will contact you if anything needs to be revised before copyediting and production can begin. Otherwise, you will be notified when your proofs are ready to be viewed.

Sincerely,
Sadjia Bekal
Editor
Microbiology Spectrum